# Factors associated with HIV testing among pregnant women in Rwanda: A nationwide cross-sectional survey

**Lilian Nuwabaine**[1]*, **Joseph Kawuki**[2], **Angella Namulema**[3], **John Baptist Asiimwe**[1], **Quraish Sserwanja**[4], **Ghislaine Gatasi**[5], **Elorm Donkor**[2]

**1** School of Nursing and Midwifery, Aga Khan University, Kampala, Uganda, **2** Jockey Club School of Public Health and Primary Care, Faculty of Medicine, The Chinese University of Hong Kong, Hong Kong, SAR-China, **3** Mbarara Regional Referral Hospital, Mbarara, Uganda, **4** Programmes Department, Relief International, Khartoum, Sudan, **5** Key Laboratory of Environmental Medicine Engineering, School of Public Health, Southeast University, Nanjing, Jiangsu Province, China

* lilliannuwabaine@gmail.com

**Data Availability Statement:** The data set used is openly available upon permission from the

## Abstract

Human immunodeficiency virus (HIV) testing during pregnancy is crucial for the prevention of mother-to-child transmission of HIV, through aiding prompt treatment, care, and support. However, few studies have explored HIV testing among pregnant women in Rwanda. This study, therefore, aimed to determine the prevalence and associated factors of HIV testing among pregnant women in Rwanda. We used secondary data from the 2020 Rwanda Demographic and Health Survey (RDHS), comprising 870 pregnant women. Multistage stratified sampling was used by the RDHS team to select participants. We conducted bivariable and multivariable logistic regression to explore factors associated with HIV testing using SPSS (version 25). Of the 870 pregnant women, 94.0% had tested for HIV during their current pregnancy. Younger age (24–34 years), not working, large household size, multiple sex partners, as well as secondary, primary, and no education were associated with higher odds of HIV testing compared to their respective counterparts. However, being unmarried, belonging to the western region, having not visited a health facility, and not having comprehensive HIV knowledge were associated with lower odds of HIV testing. A high proportion of pregnant women had tested for HIV. The study revealed that individual-level factors had the greatest influence on HIV testing in pregnancy, with a few household-level factors showing significance. There is a need for maternal health stakeholders to design and develop HIV testing programs that are region-sensitive. These programs should target older, more educated, working, and unmarried women with limited HIV knowledge.

## Background

The Human Immunodeficiency Virus (HIV) remains a significant public health problem, with women accounting for the largest HIV disease burden [1, 2]. In 2021 alone, 54% of the 38.4 million people living with HIV (PLWH) worldwide were women and girls, with pregnant

MEASURE DHS website (URL: https://www.dhsprogram.com/data/available-datasets.cfm). However, authors are not authorized to share this data set with the public but anyone interested in the data set can seek it with written permission from the MEASURE DHS website (URL: https://www.dhsprogram.com/data/available-datasets.cfm).

**Funding:** The authors received no specific funding for this work.

**Competing interests:** The authors have declared that no competing interests exist.

women inclusive, and 15% of these were not aware of their HIV status [1]. In sub-Saharan Africa, about half of the 33 million PLWH were women of reproductive age (15–49 years), with HIV rates ranging from 15–40% among pregnant women [3, 4].

HIV testing during pregnancy is crucial in preventing child-maternal transmission of HIV, as it enables timely detection of HIV, rapid initiation of treatment, and support during and after childbirth [5]. For pregnant women, testing and knowing their HIV status helps early linkage to care for those diagnosed with HIV, and this reduces the risk of transmitting HIV to the fetus and newborn [6]. According to Osório et al. [7], mother-to-child transmission (MTCT) of HIV is the main mode of HIV transmission in children under the age of 15 years. Over 80% of children living with HIV live in sub-Saharan Africa where this problem is significant [7].

In sub-Saharan Africa, over 180,000 children are infected with HIV during pregnancy, childbirth, and breastfeeding by mothers aged 15–49 [8–10]. Several studies have reported a great risk of MTCT of HIV during pregnancy ranging from 15–45% without antiretroviral therapy (ART), but this can be reduced to less than 5% with prenatal testing of pregnant women for HIV and early linkage to HIV care [11–13]. In sub-Saharan Africa, the rate of MTCT of HIV decreased from 27.2% in 2010 to 16.9% in 2019, which decline is largely attributed to early detection and linkage to HIV care during antenatal care (ANC) [14]. However, the rate of HIV testing among pregnant women is still very low in most sub-Saharan African countries [15].

Several factors increase pregnant women's susceptibility to HIV acquisition and its impact, including biological, behavioral, socioeconomic, cultural, and structural risks [16]. In most African countries, there are high rates of sexism where men dominate in cultural and societal affairs, which aggravates women's marginalization, and in most cases, women do not play a role in sexual decision-making [17]. Besides, high rates of poverty, gender-based violence, and inequality collectively pose a high risk for HIV infection among pregnant women in Africa [18, 19]. This vulnerability further solidifies the importance of prompt HIV testing among pregnant women. Studies conducted among adolescent girls and young women in Rwanda and other sub-Saharan African countries reported that failure to acquire permission from the husband, fear of stigma and discrimination, perceived lack of secrecy from service providers, and inability to access the health facility hindered pregnant women from testing for HIV [9, 20].

The government of Rwanda, through the Ministry of Health (MoH), adopted strategies to promote HIV testing services among pregnant women through the National Strategic Plan (NSP) of July 2013-June 2018, including mandatory HIV testing during antenatal care (ANC), community-based testing, self-testing, and door-to-door testing [20, 21]. The National Strategic Plan aimed at zero new HIV infections, zero HIV-related deaths, and zero stigma and discrimination due to HIV [22]. Although HIV testing remains a common component of ANC, only 13.1% of women in Rwanda receive all the components of ANC [23, 24]. Indeed, 47.2% of pregnant women in Rwanda utilize at least four ANC contacts, which limits the opportunities for access to HIV testing services for pregnant women [23, 24].

Previous studies in Rwanda evaluated HIV testing among the general population and other groups such as the male population [20, 25], with limited accessible studies focusing on pregnant women. Consequently, the level of utilization of HIV testing services and associated factors among pregnant women in Rwanda remains unclear. Previous studies elsewhere showed that pregnant women with a higher educational level, married, high wealth index, living in urban areas, and being exposed to media had higher rates of HIV testing [26–29]. These factors were considered in the present study to evaluate their relevance among Rwandan pregnant women.

Therefore, this study aimed to determine the prevalence and factors associated with HIV testing among pregnant women in Rwanda using the most recent 2020 Rwanda Demographic

and Health Survey. Understanding the factors associated with HIV testing among this specific group is crucial in addressing the reasons for the unmet need for HIV testing in Rwanda and other sub-Saharan African countries.

## Methods

### Study sampling and participants

This was a secondary data analysis of the 2019–20 Rwanda Demographic Health Survey (RDHS), a cross-sectional survey. The RDHS employed a two-stage sample design, with the first stage involving cluster selection consisting of enumeration areas and the second stage involving systematic sampling of households in all the selected enumeration areas, leading to a total of 13,005 households [30]. The data used in this analysis were from household and women's questionnaires.

The RDHS data collection period was from November 2019 to July 2020, and eligible women for the interviews were those aged 15–49 years, including permanent residents of the selected households or visitors who stayed in the household the night before the survey [30]. Out of the total 13,005 households that were selected for the survey, 12,951 were occupied, and 12,949 were successfully interviewed, leading to a 99.9% response rate [30]. For this analysis, we included only pregnant women interviewed during the survey, which were 870 out of the total 14,634 women in the whole survey [30]. During the RDHS, women were asked "Are you currently pregnant?", and this enabled us to filter out only pregnant women at the time of the survey. Given the study scope, we decided to exclude women who had recently given birth to ensure that our findings and recommendations are specific to pregnant women.

### Variables

**Dependent variables.** The outcome variable was the uptake of HIV testing during pregnancy, which was a binary outcome variable coded as yes or no and was self-reported [30].

**Explanatory variables.** We included possible determinants of HIV testing based on the available literature and data [10, 20, 30, 31]. These included: place of residence (categorized into rural and urban), region of residence (Kigali, South, West, East, North), perceived problems of distance from the household to the nearby health facility (big problem, no big problem), household size (less than six, six and above), sex of household head (male, female), husband/partner's educational level, wealth index (categorized into five quintiles that ranged from the poorest to the richest quintile), age (15–24, 25–34, 35 and above), working status (yes, no), parity (number of children ever born (4 and less, above 4) educational level (no education, primary, secondary, tertiary), marital status (married, unmarried), health insurance (yes, no), visited a health facility in last 6 months (yes, no), exposure to mass media (yes, no), having multiple sex partners (yes, no), history of sexually transmitted infections (STIs) (yes, no), healthcare decision-making (yes, no), knowledge of HIV testing kits (yes, no), comprehensive knowledge of HIV and MTCT (both coded as yes, no) and HIV stigma (no stigma, low, high). The wealth index was calculated by RDHS from information on household asset ownership using principal component analysis [30]. Health insurance was considered if a respondent had any type of insurance that covered the whole or a part of the risk incurred from medical expenses and included both public and private insurance [30]. Media exposure was measured as a woman having access to any of these: radio, newspapers, and television. Comprehensive knowledge of HIV was assessed using a six-item questionnaire: three on HIV prevention information and three questions on misconceptions of HIV transmission modes. Comprehensive knowledge of the MTCT of HIV was confirmed when a woman reported knowing about the possibility of HIV transmission from an HIV-positive mother to her child during pregnancy, delivery, and breastfeeding (that is, scoring all three questions correctly).

Moreover, the HIV stigma index was computed from six questions reflecting negative attitudes towards people living with HIV/AIDS. This index was graded "no stigma" (score 0), "low stigma" (score 1–3), and "high stigma" (score 4–6) [8, 20].

## Statistical analysis

Demographic and Health Survey (DHS) sample weights were applied to account for the unequal probability sampling in different strata and ensure the representativeness of the study results [32, 33]. The Statistical Package for Social Sciences (SPSS) (version 25.0)-complex samples package was used, incorporating the following variables in the analysis plan to account for the multistage sample design inherent in the RDHS dataset: individual sample weight, sample strata for sampling errors/design, and cluster number [32, 33].

We used frequency distributions to describe the background characteristics of the respondents, where frequencies and proportions/percentages for categorical dependent and independent variables were presented. We then conducted bivariable logistic regression to assess the association of each predictor variable with the outcome variable (HIV testing), and we presented the respective crude odds ratio (COR), 95% confidence interval (CI), and p-values. Independent variables found significant at a p-value $<0.25$ were then included in the multivariable model, including those reported to be significantly associated with HIV testing in previous studies, regardless of their significance on bivariable analysis. In other words, all variables in bivariable analysis were included in the multivariable logistic regression model. This model approach was used to avoid the possible elimination of factors that, although non-significant in bivariable analysis, may become significant when adjusted for other variables in the model [34]. Backward elimination was then executed until the model contained only significant variables with p-value $<0.05$. Respective adjusted odds ratios (AOR), 95%CI, and p-values were obtained and presented, with a statistical significance level set at p-value $<0.05$. All predictor variables in the model were assessed for multi-collinearity, which was considered present if a variable had a variance inflation factor (VIF) greater than 10 [35]. However, none of the variables had a VIF above 3. Model fitness was assessed with the Hosmer-Lemeshow test, where small Chi-squared values (with a larger p-value closer to 1) implied significant model fitness [36]. Missing data in explanatory variables were handled by a list-wise deletion in SPSS.

## Ethical considerations

High international ethical standards are ensured during MEASURE DHS surveys, and the study protocol is performed following the relevant guidelines. The RDHS 2019 survey protocol was reviewed and approved by the Rwanda National Ethics Committee (RNEC) and the ICF Institutional Review Board. RDHS data collectors obtained written informed consent from human participants, and written informed consent was also obtained from legally authorized representatives of minor participants. Since this was a secondary data analysis of a publicly available dataset, additional ethical approval was waived by RDHS. The RDHS data set was obtained from the MEASURE DHS website (URL: https://www.dhsprogram.com/data/available-datasets.cfm) after getting their written permission. Researchers for this project are not affiliated with the DHS team for this survey.

## Results

### Characteristics of participants

A total of 870 pregnant women were included in this analysis (**Table 1**). The majority were below 35 years of age (74.2%), married (84.7%), had attained a primary level of education (60%), were working (66.1%), and were rural residents (81.5%). In addition, most of the

**Table 1. Background characteristics of pregnant women as per the 2020 Rwanda demographic health survey.**

| Characteristics | Frequency (%), N = 870 |
|---|---|
| **Age (years old)** | |
| 15–24 | 245(28.2) |
| 25–34 | 400(46.0) |
| 35 and above | 225(25.8) |
| **Education level** | |
| No education | 74(8.6) |
| Primary | 522(60.0) |
| Secondary | 230(26.4) |
| Tertiary | 44(5.0) |
| **Working status** | |
| Working | 575(66.1) |
| Not working | 295(33.9) |
| **Number of children ever born** | |
| Below 4 | 724(83.3) |
| 4 and above | 146(16.7) |
| **Marital status** | |
| Married | 737(84.7) |
| Unmarried | 133(15.3) |
| **Health insurance** | |
| Yes | 775(89.1) |
| No | 95(10.9) |
| **Wealth index** | |
| Richest | 190(21.8) |
| Richer | 173(19.9) |
| Middle | 178(20.5) |
| Poorer | 175(20.1) |
| Poorest | 154(17.7) |
| **Residence** | |
| Urban | 161(18.5) |
| Rural | 709(81.5) |
| **Region** | |
| Kigali | 110(12.6) |
| West | 199(22.9) |
| East | 222(25.5) |
| North | 131(15.1) |
| South | 208(23.9) |
| **Household size** | |
| Less than 6 | 676(77.7) |
| 6 and above | 194(22.3) |
| **Sex of household head** | |
| Male | 714(82.0) |
| Female | 156(18.0) |
| **Husband's education [a]** | |
| Tertiary | 62(7.2) |
| Secondary | 120(13.9) |
| Primary | 460(52.9) |
| No education | 94(10.8) |

(*Continued*)

**Table 1.** (Continued)

| Characteristics | Frequency (%), N = 870 |
|---|---|
| **Exposure to media** | |
| Yes | 719(82.6) |
| No | 151(17.4) |
| **Perceived problems of distance from the household to the nearby health facility** | |
| No big problem | 660(75.8) |
| Big problem | 210(24.2) |
| **Visited health facility in the last 6 months** | |
| Yes | 734(84.4) |
| No | 136(15.6) |
| **Multiple sex partners** | |
| Yes | 118(13.6) |
| No | 752(86.4) |
| **STI history** | |
| Yes | 47(5.4) |
| No | 823(94.6) |
| **Healthcare decision-making [b]** | |
| Yes | 583(67.0) |
| No | 154(17.7) |
| **Comprehensive MTCT knowledge** | |
| Yes | 595(68.4) |
| No | 275(31.6) |
| **Comprehensive HIV knowledge** | |
| Yes | 599(68.8) |
| No | 271(31.2) |
| **Know about HIV test kits** | |
| Yes | 147(16.9) |
| No | 723(83.1) |
| **HIV-related stigma** | |
| No | 85(9.7) |
| Low | 672(77.3) |
| High | 113(13.0) |
| **Ever tested for HIV** | |
| No | 52(6.0) |
| Yes | 818(94.0) |

Missing values a = 134, b = 134: Husband's education and Healthcare decision-making were both collected by the domestic violence questionnaire and thus have the same number of missing values. MTCT = Mother-to-child transmission, HIV = Human immune virus, STI = Sexually transmitted diseases

participants had ever given birth to less than four children (83.3%), had health insurance (81.9%), were from a household size of fewer than six members (77.7%), of which the majority were headed by males (82%), had exposure to media (82.6%), had no problem accessing the health facility (75.8%), and had visited a health facility in the last six months (84.4%). Furthermore, the majority had no multiple-sex partners (86.4%), no history of STIs (94.6%), participated in healthcare decision-making (67%), had comprehensive HIV knowledge (68%) and MTCT knowledge (68.4%), and had low HIV-related stigma (77.3%). Regarding HIV testing, 94.0% (95%CI: 92.3–95.4) had ever tested for HIV during pregnancy.

### Factors associated with HIV testing among pregnant women

Results of the bivariable analysis are detailed in **Table 2,** with significant factors independently associated with HIV testing highlighted. After adjusting for all the included variables in the multivariable logistic regression model, the factors found to have a significant association included age, education level, working status, marital status, region, household size, visit to a health facility in the last 6 months, multiple sexual partners, and comprehensive HIV knowledge (**Table 2, see AOR**).

Pregnant women aged 25–34 years were 1.54(95%CI: 1.54–4.42) times more likely to test for HIV compared to those aged 35 and above. Compared to those with tertiary education, pregnant women with secondary, primary, and no education were 8.07(95%CI: 2.15–11.43), 5.53(95%CI: 1.28–9.74), and 6.07(95%CI: 1.21–10.44) times, respectively, more likely to test for HIV. Similarly, compared to working pregnant women, those not working were 4.29(95% CI: 1.52–12.08) times more likely to test for HIV. Those from households of 6 members and above were also 2.96(95%CI: 1.01–8.61) times more likely to test for HIV compared to their counterparts from households of less than 6 members. Moreover, pregnant women with no multiple sex partners were 4.16(95%CI: 3.01–5.74) times more likely to test for HIV compared to those with multiple sex partners.

However, unmarried women (AOR = 0.28, 95%CI: 0.19–0.86) and those from the western region (AOR = 0.20, 95%CI: 0.63–0.66) were 0.28 and 0.20 times less likely to test for HIV, compared to married pregnant women and those from Kigali, respectively. Similarly, pregnant women who had not visited a health facility in the last 6 months (AOR = 0.22, 95%CI: 0.10–0.48) and those with no comprehensive HIV knowledge (AOR = 0.68, 95%CI: 0.30–0.55) were 0.22 and 0.68 times less likely to test for HIV compared to those that had visited a health facility and had comprehensive HIV knowledge, respectively.

### Discussion

In this study, the prevalence and factors associated with HIV testing among pregnant women in Rwanda were assessed. The study identified potential factors influencing HIV testing among pregnant women and these have a great impact on both mother and baby if not addressed. The prevalence of HIV testing among pregnant women was found to be 94.0%. This is higher than the prevalence of HIV testing in Zambia (80%) among women of childbearing age and the overall prevalence reported from 11 East African countries (67.13%) and 85.74% specifically for Rwanda [27, 29]. The variations in quality and access to HIV testing facilities, as well as knowledge related to HIV/AIDS, including mandatory HIV testing during ANC, community-based testing, self-testing, and door-to-door testing, may be the reasons for the reported variations [21, 26, 29]. Additionally, the observed difference in HIV testing among pregnant women may be attributed to the differential efforts and resources put into HIV prevention programs, as well as differences in socio-demographics across countries [10, 37].

Pregnant women aged 25–34 years were more likely to test for HIV compared to their older counterparts (35 years and above). This could be attributed to the fact that sexual education tends to be focused on young women who may be involved in high-risk sexual behaviors and that older women do not perceive themselves to be at risk for HIV [27, 38, 39]. These findings are consistent with studies from Vietnam, Ethiopia, and 11 East African countries, which also reported older age to be associated with less HIV testing [40, 41]. Nonetheless, further studies should be conducted to critically identify reasons why older women (35 years and above) are less likely to test for HIV during pregnancy in Rwanda.

**Table 2.** Factors associated with HIV testing among pregnant women, as per the 2020 Rwanda demographic health survey.

| Variable | Distribution of HIV testing, N = 818, n (Row %) | Crude odds ratio, COR (95% CI) | p-value* | Adjusted odds ratio, AOR (95% CI) | p-value** |
|---|---|---|---|---|---|
| **Age (years old)** | | | < 0.001 | | **0.015** |
| 35 and above | 215(96.0) | 1 | | 1 | |
| 25–34 | 387(96.5) | 1.22(1.53–2.84) | | **1.54(1.54–4.42)** | |
| 15–24 | 216(87.8) | 0.31(0.15–0.66) | | 0.40(0.12–1.32) | |
| **Education level** | | | 0.443 | | **0.046** |
| Tertiary | 42(95.5) | 1 | | 1 | |
| Secondary | 221(96.1) | 1.01(0.22–4.66) | | **8.07(2.15–11.43)** | |
| Primary | 485(92.9) | 0.52(0.12–2.38) | | **5.53(1.28–9.74)** | |
| No education | 70(94.6) | 0.64(0.11–3.61) | | **6.07(1.21–10.44)** | |
| **Working status** | | | 0.030 | | **0.006** |
| Working | 533(92.5) | 1 | | 1 | |
| Not working | 285(96.6) | 2.36(1.09–5.12) | | **4.29(1.52–12.08)** | |
| **Number of children ever born** | | | 0.475 | | 0.400 |
| 4 and above | 140(95.9) | 1 | | 1 | |
| Below 4 | 678(93.5) | 0.61(0.23–1.61) | | 1.89(0.43–8.35) | |
| **Marital status** | | | 0.017 | | **0.021** |
| Married | 699(94.8) | 1 | | 1 | |
| Unmarried | 119(89.5) | 0.45(0.23–0.87) | | **0.28(0.19–0.86)** | |
| **Health insurance** | | | 0.144 | | 0.914 |
| Yes | 732(94.4) | 1 | | 1 | |
| No | 86(90.5) | 0.56(0.25–1.22) | | 0.93(0.26–3.29) | |
| **Wealth index** | | | 0.127 | | 0.655 |
| Richest | 183(96.3) | 1 | | 1 | |
| Richer | 167(96.0) | 0.91(0.35–2.37) | | 1.56(0.38–6.36) | |
| Middle | 169(94.9) | 0.69(0.26–1.84) | | 2.26(0.58–8.88) | |
| Poorer | 161(92.5) | 0.46(0.18–1.20) | | 2.98(0.71–12.42) | |
| Poorest | 138(89.6) | 0.33(0.13–0.85) | | 2.16(0.49–9.47) | |
| **Residence** | | | 0.129 | | 0.506 |
| Urban | 157(97.5) | 1 | | 1 | |
| Rural | 661(93.1) | 0.34(0.08–1.38) | | 0.56(0.10–3.08) | |
| **Region** | | | < 0.001 | | **< 0.001** |
| Kigali | 110(100.0) | 1 | | 1 | |
| West | 180(89.9) | 0.11(0.62–0.19) | | **0.20(0.63–0.66)** | |
| East | 210(94.6) | 0.21(0.97–4.57) | | 0.89(0.19–4.19) | |
| North | 123(93.9) | 0.19(0.90–4.06) | | 0.78(0.17–3.46) | |
| South | 195(93.8) | 0.18(0.11–3.20) | | 0.83(0.24–2.85) | |
| **Household size** | | | 0.287 | | **0.047** |
| Less than 6 | 633(93.5) | 1 | | 1 | |
| 6 and above | 185(95.4) | 1.50(0.71–3.15) | | **2.96(1.01–8.61)** | |
| **Sex of household head** | | | 0.830 | | 0.719 |
| Male | 671(93.8) | 1 | | 1 | |
| Female | 147(94.2) | 1.09(0.51–2.30) | | 1.27(0.35–4.59) | |
| **Husband's education** | | | 0.510 | | 0.558 |
| Tertiary | 61(98.4) | 1 | | 1 | |
| Secondary | 116(95.9) | 0.39(0.04–3.50) | | 0.24(0.03–2.27) | |
| Primary | 433(94.1) | 0.26(0.03–2.01) | | 0.20(0.02–1.81) | |

*(Continued)*

**Table 2.** (Continued)

| Variable | Distribution of HIV testing, N = 818, n (Row %) | Crude odds ratio, COR (95% CI) | p-value* | Adjusted odds ratio, AOR (95% CI) | p-value** |
|---|---|---|---|---|---|
| No education | 88(94.6) | 0.27(0.03–2.33) | | 0.20(0.02–2.26) | |
| **Exposure to media** | | | 0.183 | | 0.446 |
| Yes | 680(94.4) | 1 | | 1 | |
| No | 138(91.4) | 0.63(0.32–1.24) | | 1.45(0.56–3.74) | |
| **Perceived problems of distance from the household to the nearby health facility** | | | 0.589 | | 0.442 |
| No big problem | 618(93.6) | 1 | | 1 | |
| Big problem | 200(94.8) | 1.23(0.59–2.56) | | 1.44(0.57–3.66) | |
| **Visited health facility in last 6 months** | | | < 0.001 | | **< 0.001** |
| Yes | 704(95.9) | 1 | | 1 | |
| No | 114(83.2) | 0.22(0.12–0.40) | | **0.22(0.10–0.48)** | |
| **Multiple sex partners** | | | 0.100 | | **0.030** |
| Yes | 107(90.7) | 1 | | 1 | |
| No | 711(94.5) | 1.84(0.89–3.83) | | **4.16(3.01–5.74)** | |
| **STI history** | | | 0.714 | | 0.409 |
| Yes | 44(95.7) | 1 | | 1 | |
| No | 774(93.9) | 0.76(0.17–3.34) | | 1.79(0.45–7.11) | |
| **Healthcare decision-making** | | | 0.287 | | 0.957 |
| Yes | 555(95.4) | 1 | | 1 | |
| No | 143(92.9) | 0.65(0.31–1.33) | | 0.97(0.36–2.64) | |
| **Comprehensive MTCT knowledge** | | | 0.123 | | 0.457 |
| Yes | 555(93.1) | 1 | | 1 | |
| No | 263(95.6) | 1.66(0.87–3.17) | | 1.39(0.58–3.30) | |
| **Comprehensive HIV knowledge** | | | 0.041 | | **0.035** |
| Yes | 571(95.3) | 1 | | 1 | |
| No | 247(91.1) | 0.51(0.26–0.97) | | **0.68(0.30–0.55)** | |
| **Know about HIV test kits** | | | 0.250 | | 0.786 |
| Yes | 141(95.9) | 1 | | 1 | |
| No | 677(93.5) | 0.61(0.27–1.41) | | 0.85(0.26–2.74) | |
| **HIV-related stigma** | | | 0.360 | | 0.355 |
| No | 83(97.6) | 1 | | 1 | |
| Low | 631(93.9) | 0.44(0.11–1.77) | | 0.25(0.03–1.81) | |
| High | 104(92.0) | 0.32(0.07–1.55) | | 0.20(0.02–1.89) | |

**Bold** = significant adjusted factors, * = significant at 0.25, ** = significant at 0.05, CI = confidence interval, RDHS = Rwanda demographic health survey,

MTCT = mother-to-child transmission, HIV = Human immune virus, STI = sexually transmitted diseases

The study results showed that pregnant women who lived in the western region of the country had lower odds of HIV testing compared to those who lived in Kigali. The study findings are supported by other studies in Rwanda and Senegal which also reported differential rates in HIV testing among regions [20, 21]. Kigali as the capital city of Rwanda and predominantly urban has more widespread and multi-service health infrastructures which offer more opportunities for its residents to access diverse health services. The western region of Rwanda, on the other hand, is 87.8% rural and relatively remote, making it less developed with fewer health facilities, and therefore lesser chances to access health services including sexual and reproductive awareness [25]. Further, compared to Kigali, rural Rwanda might have less

variety of media exposure in the form of television, radio, billboards, and social media which are key sources of information about the importance of HIV testing during pregnancy [25, 42]. It is therefore imperative to ensure balanced efforts and resource allocation in HIV prevention programs across the country. Moreover, developing an appropriate and targeted community-based intervention such as community or home visits by health workers to test individuals in remote and hard-to-reach areas would increase access to HIV testing services among pregnant women [11].

Surprisingly, lower levels of education (secondary, primary, and none) were associated with higher odds of HIV testing compared to tertiary level of education. Notably, tertiary education was associated with fewer odds of HIV testing, a finding contrary to previous studies associating higher educational attainment with a higher likelihood of HIV testing [29, 31, 43]. Education is well established to be a potent tool that may be utilized to not only improve HIV knowledge but also to empower individuals, especially women, to make the right decisions regarding attendance and utilization of health services [44, 45]. However, the probable reason for the lower odds of HIV testing among pregnant women with tertiary education could be attributed to the stigmatizing attitudes more educated people tend to hold towards HIV-positive people; hence, they may fear utilizing HIV testing services [46]. In addition, it is argued that more educated women are likely to have demanding careers with tight work schedules and thus less free time for ANC/HIV testing. The actual reason remains inconclusive and therefore warrants further exploration to establish why more educated women were less likely to use HIV testing services.

Pregnant women in this study with no comprehensive knowledge about HIV were less likely to test for HIV compared to those with comprehensive knowledge. The study finding is consistent with a study among women in East Africa which reported high HIV education was associated with HIV testing [27]. This is not surprising and further highlights a need to strengthen the means of HIV information dissemination through the use of media campaigns and emphasis on HIV health education and counseling by health workers during antenatal care and community outreaches. This should consider addressing possible misconceptions among various education levels and needs.

Furthermore, it was observed that pregnant women who had visited a health facility in the last 6 months had higher odds of HIV testing compared to their counterparts who had not. This finding is supported by a similar study in Ethiopia which also reported a positive association between recent health facility visits and HIV testing [12]. Pregnant women who visit a health facility are more likely to attend antenatal care, of which health education and HIV testing are compulsory components [37], explaining the above observation. Pregnant women from large household sizes were also more likely to be tested for HIV compared to their counterparts from smaller households. This is in agreement with a study from Sudan, which also reported that women in extended families are more likely to test for HIV [47]. This may be because of the influence and concern of their older relatives, who may tend to encourage them to attend antenatal care where they can access and receive HIV testing services.

Married women were also more likely to test for HIV during pregnancy compared to unmarried. This is similar to a study from Thailand [37] and that from East Africa [31] which also reported a positive association between marital status and HIV testing. The possible explanation may be that married women are more likely to be reminded and encouraged by their partners to go for antenatal care where they can receive HIV testing services [48, 49]. Moreover, since married women tend to have premarital HIV testing with their partners, they, thus, tend to be more receptive to regular HIV testing [48, 49]. Additionally, it was also observed that pregnant women without multiple sexual partners were more likely to test for HIV compared to those with multiple sexual partners. This is in agreement with a study conducted in

East Africa, which showed that women with multiple sexual partners had lower odds of HIV testing [31]. On the other hand, due to differences in the context, studies in Ethiopia identified women with multiple sexual partners as having higher odds of HIV testing compared to those with one sexual partner, due to a higher perceived risk of acquiring HIV [12, 50]. Nonetheless, further studies are needed to explore the association between the number of sexual partners and HIV testing during pregnancy.

Study findings also indicate that non-working pregnant women had higher odds of HIV testing compared to those working. The study finding is consistent with a study in Vietnam which reported employed pregnant women being more likely to test for HIV [41]. Although employment is a vital component of women's empowerment for making healthy decisions and choices, it has also been reported to harm other women's health aspects, predisposing women to sexual violence [51] and reducing antenatal and postnatal care attendance [52, 53] partly due to the busy work schedules. This implies a need to strengthen work-related policies, such as work leaves, to enable working women to attend to vital health aspects of pregnancy and child care without risking their jobs and careers.

## Strengths and limitations

We used the most recent data from the 2020 Rwanda Demographic and Health Survey, with a relatively large sample size and standardized data collection protocols. Moreover, we weighted the dataset, making our results generalizable to all pregnant women in Rwanda. However, this study also had some limitations, such as the risk of missing information and recall bias due to the sensitivity of the HIV topic (prone to social desirability effects), and the data was self-reported without verifying with health records. All this could have affected the true estimation of HIV testing prevalence. The cross-sectional design of this study also limits inferring causality, but rather only association. Therefore, the interpretation of odds ratios in this study should be done with caution given the cross-sectional nature of the study, so they should be framed in terms of odds, not in terms of probabilities since they do not imply relative risk. Moreover, the secondary data used in this study could not fully explore all potential determinants of HIV testing in pregnancy and could not explain the underlying reasons. We therefore recommend future qualitative studies to fully examine the underlying reasons for failure to take HIV testing among pregnant women in the context of Rwanda. Moreover, the logistic regression model used in our analysis assumes linearity between the predictor and the outcome variable, which may not be the case in real-world settings. Some of the predictors of HIV testing may be mediated by other factors, thus future studies on the topic should consider using more rigorous analytical methods such as mediation analysis and structural equation modeling.

## Conclusions and public health implications

The study revealed that a large proportion of pregnant women in Rwanda had tested for HIV during pregnancy, and the proportion was the highest in the East African region. The study also revealed that individual-level factors had the greatest influence on HIV testing in pregnancy (age, marital status, education level, working status, comprehensive HIV knowledge, visit to a health facility, and having multiple sex partners), with only two household-level factors showing significance (region and household size). The identified factors emphasize the significance of social determinants of health as well as the need for programs/interventions that focus beyond improving physical access since the identified predictors act on both the supply and demand sides of HIV testing services. Implementation of proven interventions to improve access and uptake of HIV testing among pregnant women should be tailored according to geographical locations and household size at the community/household level, and at the

individual level, older, more educated, working women with low HIV knowledge and few sex partners should be targeted. The results also imply a need for improved access to HIV education and testing facilities for pregnant women to address the regional imbalances, for example, through massive community awareness about HIV via campaigns and outreaches. Furthermore, there is a need to explore why highly educated and working women are less likely to test for HIV and set up favorable policies that enable them to easily attend antenatal/HIV testing services. Such policies can include giving pregnant women flexible working hours with less tight work schedules.

## Supporting information

**S1 Checklist.**
(DOCX)

## Author Contributions

**Conceptualization:** Lilian Nuwabaine, Joseph Kawuki.

**Data curation:** Lilian Nuwabaine, Joseph Kawuki.

**Formal analysis:** Lilian Nuwabaine, Joseph Kawuki.

**Investigation:** Lilian Nuwabaine.

**Methodology:** Lilian Nuwabaine, Joseph Kawuki.

**Project administration:** Lilian Nuwabaine.

**Software:** Joseph Kawuki.

**Supervision:** Lilian Nuwabaine.

**Writing – original draft:** Lilian Nuwabaine, Joseph Kawuki, Angella Namulema.

**Writing – review & editing:** Lilian Nuwabaine, Joseph Kawuki, Angella Namulema, John Baptist Asiimwe, Quraish Sserwanja, Ghislaine Gatasi, Elorm Donkor.

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
