## [Decision Letter · Decision Letter 0]

2 Aug 2023

PGPH-D-23-00992

Human Immunodeficiency Virus testing and associated factors among pregnant women in Rwanda: a nationwide cross-sectional survey

Dear Mrs. Lilian Nuwabaine,

Thank you for submitting your manuscript to PLOS Global Public Health. After careful consideration, we feel that it has merit but does not fully meet PLOS Global Public Health’s publication criteria as it currently stands. Therefore, we invite you to submit a revised version of the manuscript that addresses the points raised during the review process.

We look forward to receiving your revised manuscript.

Kind regards,

Muriel Mac-Seing, PhD

Academic Editor

 Journal Requirements:

1. Your manuscript is missing the following sections: Introduction. Please ensure these are present, and in the correct order, and that any references to subheadings in your main text are correct. An outline of the required sections can be consulted in our submission guidelines here:

https://journals.plos.org/globalpublichealth/s/submission-guidelines#loc-parts-of-a-submission

2. We have noticed that you have uploaded Supporting Information files, but you have not included a list of legends. Please add a full list of legends for your Supporting Information files after the references list.

Reviewers' comments:

Reviewer's Responses to Questions

**Comments to the Author**

1. Does this manuscript meet PLOS Global Public Health’s publication criteria? Is the manuscript technically sound, and do the data support the conclusions? The manuscript must describe methodologically and ethically rigorous research with conclusions that are appropriately drawn based on the data presented.

Reviewer #1: Partly

Reviewer #2: Yes

2. Has the statistical analysis been performed appropriately and rigorously?

Reviewer #1: No

Reviewer #2: Yes

3. Have the authors made all data underlying the findings in their manuscript fully available (please refer to the Data Availability Statement at the start of the manuscript PDF file)?

Reviewer #1: Yes

Reviewer #2: Yes

4. Is the manuscript presented in an intelligible fashion and written in standard English?

Reviewer #1: Yes

Reviewer #2: Yes

5. Review Comments to the Author

Reviewer #1: Summary of the research and overall impression:

First, I want to thank the authors for this article that hopes to address some important gaps in understanding the factors related to HIV testing in pregnant women in Rwanda. The use of data from the Rwanda Demographic and Health Survey for this secondary analysis provides valuable insights into the frequency and associations of factors related to HIV testing. However, after a careful review of the article, I believe that it requires a major revision. The writing style and presentation of the paper need improvement to enhance its clarity and coherence. Additionally, the statistical soundness of the paper should be carefully addressed to ensure the validity of the findings and the robustness of the conclusions.

Discussion of specific areas for improvement

- Clarity and Writing Style: The article would benefit from a more concise and clear writing style. Complex sentences and overly technical language may hinder the readers' comprehension of the study's objectives and findings. Simplifying the language and providing clearer explanations of statistical methods would enhance the paper's accessibility.

- Statistical Analysis: While the paper presents associations between various factors and HIV testing, the statistical methodologies employed need to be rigorously reviewed. Please ensure that appropriate statistical tests and adjustments have been applied to support the validity of the results.

- Discussion of Findings: The paper should offer a more in-depth discussion of the findings, comparing them to existing literature and discussing their implications for public health in Rwanda. Emphasizing the unique contributions of this study will enrich the discussion section.

- Suggestions for Future Research: Consider including suggestions for future research to address any remaining gaps or explore related aspects of HIV testing among pregnant women. Proposing areas for further investigation will add value to the article.

More Specific Comments:

- Consider revising the title to use the acronym for HIV, which would make it more concise and familiar to readers.

- The abstract should be improved to enhance readability and clarity. The excessive use of AORs makes it challenging to grasp the main results. Consider rephrasing to present the key findings more succinctly and clearly.

- Abstract conclusion: Provide more context in the conclusion about the implications of the associations and factors found in the study. Discuss their potential significance and relevance to public health and policy in Rwanda.

- Introduction, line 52-53: Clarify why women yield the largest burden of HIV in line 52-53. Provide a contextual explanation of the factors contributing to the higher prevalence of HIV in women in sub-Saharan Africa. Also, verify the appropriateness of reference 2.

- Introduction, first paragraph: Explicitly state why pregnant women matter regarding HIV testing. Highlight the significance of understanding HIV testing among this specific group.

- Appropriateness of references: Verify the appropriateness of references used in the introduction. For example, check if reference 7 is suitable and consider using references from sub-Saharan Africa for more relevance.

- Line 63: should be integrated into the first paragraph since it is about presenting the statistics

- Sentence 65: Reframe line 65 to avoid presenting a global statistic and then moving to sub-Saharan Africa in the same sentence. Clearly present the percentage or prevalence of children who get transmitted HIV from their HIV-positive mothers.

- Line 67: Review and improve sentence structures and paragraph organization for better flow and coherence.

- Line 74: Third paragraph needs work. Provide context for the studies referenced in line 74. Explain the relevance of these studies and their findings to the current research.

- Line 79: Specify when Rwanda developed strategies to promote HIV testing services among pregnant women and the reasons behind their implementation.

- Line 82: Clarify line 82 by specifying whether it refers to all components of HIV testing or all components of ANC.

- Line 83: just need to present one statistic, the four visits since that is the common indicator of good common amount of ANC visits

- Line 85: This statement is wrong because of the articles you referenced actually does look at pregnancy as a factor. https://bmcpublichealth.biomedcentral.com/articles/10.1186/s12889-022-13679-8

- Introduction: I would like to inquire about previous statistics on HIV testing in Rwanda in general, not just among pregnant women. Additionally, it would be helpful to understand the reasons for focusing solely on women in this study. Could you provide statistics on HIV testing rates for women compared to men in Rwanda to provide better context for the rationale behind studying this specific population group? More contextual information on HIV testing trends among different demographics in Rwanda would strengthen the justification for the focus on pregnant women in this research.

- Line 90: don’t need a hyphen in “most-recent”

- Line 95: stick to one form of presentation of the objectives, you presented it in the paragraph above, don’t need to present it again below

- Introduction or method: The introduction or method section would benefit from including more contextual factors that could impact HIV testing in Rwanda. Providing information about the population size of Rwanda and the incidence of HIV in the country would help readers understand the broader context of the study. Additionally, discussing factors, both structural and social, that have been known to influence HIV testing rates in Rwanda would add valuable insights to the research. This contextual information is essential for a comprehensive understanding of the factors influencing HIV testing behavior in the specific setting of Rwanda.

- Line 119: Check your references. I don’t think reference 23 is meant to be here?

- Methodological errors should be addressed in the study. Given that the variables are described as community, household, and individual, a multilevel analysis should be considered to account for potential statistical confounding within these levels.

- Methods: What was the ethics procedure to obtain this data? Was there any procedures in place to do a secondary data analysis? Describe the ethical procedures for obtaining the data and address any ethical considerations in conducting a secondary data analysis.

- Additionally, missing data is common in DHS data. It would be valuable to provide information on whether any data was removed due to missing values and detail the procedure used for handling missing data. Transparent reporting on the treatment of missing data is essential to ensure the robustness and reliability of the study's results.

- Table 1:

o Regarding Table 1, there are some discrepancies in the numbers presented. For instance, the total number of women in the "age" category is listed as 869, which seems inconsistent with the sample size mentioned elsewhere in the article. To enhance clarity, it would be helpful to include "(years old)" next to the age variable for better understanding.

o Additionally, the education level should ideally be presented in the order of no education first, followed by primary, secondary, and tertiary. Moreover, the representation of missing values in the table should be specified for each sociodemographic variable to provide a comprehensive overview of missing data.

o Another point of concern is the identical number of missing values for both "healthcare decision making" and "husband education." It would be beneficial to clarify why these variables have the same amount of missing data, as this could potentially indicate issues with data quality or collection.

- Table 2: There are several issues with Table 2 that need to be addressed.

o Firstly, if bold formatting is used to distinguish between significant and non-significant results for adjusted analyses, the same formatting should be applied to the unadjusted analyses for consistency.

o Furthermore, the table lacks the presentation of frequencies of sociodemographic variables across the outcome variable, which is crucial for understanding the distribution and characteristics of the study population.

o Additionally, the rationale behind including all variables in the multivariable model needs clarification. It is essential to provide a clear explanation of the model building procedure and the statistical justification for including specific variables. Without this information, the soundness and validity of the multivariable model may be questioned.

- Discussion: In the discussion, provide a summary of the data's significance in the context of existing knowledge. Discuss why this data matters and suggest potential future research directions based on the study's findings.

- Limitations line 292: sentence looks like it is missing something. It is no clear what you want to mean?

- Limitations: Discuss potential limitations with the model used in the study and address any potential sources of bias.

- Clarify how the results of this study will advance research on the topic and suggest next steps or potential follow-up analyses based on the study's outcomes.Summary of the research and overall impression:

Reviewer #2: Some comments

1. I would suggest to reframe the title by emphasizing “Factors associated to HIV testing….”. This adjustment will make your title more concise and clearer. I’m suggesting this based on your study description, results and discussion.

2. Pages 3-4, lines 51-88: I would suggest you reorganize the background. Start with global, LMICs & Africa, sub-Saharan Africa and Rwanda vis-a vis the existing knowledge about HIV testing among pregnant women….. this would be helpful for a well-framed background.

3. Page 4, line 86: I suggest to do not use “no accessible study” but a different wording like “a limited number of studies, a couple of studies or few accessible studies or any other appropriate wording….” In fact, Rwanda has history of some studies that covered this topic at some extent and helped to achieve good outcomes in HTC among pregnant women such as Musekiwa et al. on prevalence and factors associated with self-reported HIV testing among adolescent girls and young women; Allen et al. on pregnancy and contraception after HIV testing and counseling; Mugwaneza et al. on ART and PMTCT; Kowalczyk et al. on voluntary counseling and testing (VCT) for HIV among pregnant women; and others.

4. Page 5, line 102: Does H in RDHS stand for Health? If so, add it.

5. For your data sampling and analysis; were you part of RDHS team? If not, how did you collaborate with them? It would be good to mention how you accessed RDHS data (permission, ethics, etc.,).

6. Page 6, lines 134-136: What was considered as comprehensive knowledge of HIV/MTCT? Unless I missed it, I didn’t see such knowledge components/items.

7. Page 8, lines 169-171: Again, specify the components/items of HIV and MTCT knowledge you were looking for.

8. Page 14, lines 225-230: This finding should have been well clarified through direct interviews instead of secondary data. Participants talking directly with researchers would have been interesting in terms of highlighting unique and contextualized factors other than those condensed in the RDHS report. Other regions in the country are rural or remote but still perform better in HTC.

9. Page 10, lines 186-188 and page 16, lines 283-285: This is a good example of how specific questions directed to participants would have unpacked or given a clear idea about different factors were associated to HTC among pregnant women. May be these findings will be a starting point for your next study using primary data!

Thanks.

6. PLOS authors have the option to publish the peer review history of their article (what does this mean?). If published, this will include your full peer review and any attached files.

**Do you want your identity to be public for this peer review?** For information about this choice, including consent withdrawal, please see our Privacy Policy.

Reviewer #1: No

Reviewer #2: No

---

## [Decision Letter · Decision Letter 1]

6 Oct 2023

PGPH-D-23-00992R1

Factors associated with HIV testing among pregnant women in Rwanda: a nationwide cross-sectional survey

Dear Dr. Nuwabaine,

Thank you for submitting your manuscript to PLOS Global Public Health. After careful consideration, one of the reviewers feels that it has merit but does not fully meet PLOS Global Public Health’s publication criteria as it currently stands. Therefore, we invite you to submit a revised version of the manuscript that addresses the points raised during the review process. Consider the reviewer's comments on how to improve the sections on sampling and your model testing. Also ensure that there are no typos left such as "Reproductive age" (P.4, Line 89) with "reproductive" that should not be capitalised, or the punctuation in "The included;" (P.7, Line 147), there should not be a semicolon. Review thoroughly your sentences.

We look forward to receiving your revised manuscript.

Kind regards,

Muriel Mac-Seing, PhD

Academic Editor

Journal Requirements:

1. Please amend your online Financial Disclosure statement. If you did not receive any funding for this study, please simply state: “The authors received no specific funding for this work.”

2. Please update your online Competing Interests statement. If you have no competing interests to declare, please state: “The authors have declared that no competing interests exist.”

Additional Editor Comments (if provided):

Reviewers' comments:

Reviewer's Responses to Questions

**Comments to the Author**

1. If the authors have adequately addressed your comments raised in a previous round of review and you feel that this manuscript is now acceptable for publication, you may indicate that here to bypass the “Comments to the Author” section, enter your conflict of interest statement in the “Confidential to Editor” section, and submit your "Accept" recommendation.

Reviewer #2: All comments have been addressed

Reviewer #3: (No Response)

2. Does this manuscript meet PLOS Global Public Health’s publication criteria? Is the manuscript technically sound, and do the data support the conclusions? The manuscript must describe methodologically and ethically rigorous research with conclusions that are appropriately drawn based on the data presented.

Reviewer #2: Yes

Reviewer #3: Partly

3. Has the statistical analysis been performed appropriately and rigorously?

Reviewer #2: I don't know

Reviewer #3: No

4. Have the authors made all data underlying the findings in their manuscript fully available (please refer to the Data Availability Statement at the start of the manuscript PDF file)?

Reviewer #2: Yes

Reviewer #3: Yes

5. Is the manuscript presented in an intelligible fashion and written in standard English?

Reviewer #2: Yes

Reviewer #3: Yes

6. Review Comments to the Author

Reviewer #2: This article has greatly improved from its last version and I would recommend its publication. It is more understandable to readers/public. If statistician/epidemiologist agrees with the presented statistical findings.

Reviewer #3: Overall Impression:

The article on the prevalence and factors associated with HIV testing among pregnant women in Rwanda provides valuable insights. The study is commendable for its comprehensive analysis and use of recent data from the 2020 Rwanda Demographic and Health Survey. The abstract and results sections are well-structured and present clear findings. However, there are several areas that could benefit from improvement to enhance the clarity and rigor of the study.

Areas for Improvement:

- The article displays commendable strengths, utilizing recent data and employing a robust sample size with standardized protocols. However, clarity is needed on the entity conducting multistage stratified sampling. The abstract could benefit from refined language. Questions arise regarding model testing methodology and variable selection criteria. Attention is required to accurate referencing and contextual integration, avoiding overgeneralizations. Structural improvements, such as breaking down lengthy paragraphs and contextual integration, can enhance readability. Ethical considerations, especially the need for institutional approval, require clarification. Variable definitions, like "parity," should be clearer, and consistency in presenting confidence intervals is essential. The discussion section would benefit from improved sentence structure. Addressing contradictory findings and emphasizing recall bias impact can strengthen the article. A more extensive discussion on study limitations, particularly the cross-sectional design, would enhance the overall quality of the manuscript.

7. PLOS authors have the option to publish the peer review history of their article (what does this mean?). If published, this will include your full peer review and any attached files.

**Do you want your identity to be public for this peer review?** For information about this choice, including consent withdrawal, please see our Privacy Policy.

Reviewer #2: No

Reviewer #3: No

---

## [Editor Report · Decision Letter 2]

7 Nov 2023

PGPH-D-23-00992R2

Factors associated with HIV testing among pregnant women in Rwanda: a nationwide cross-sectional survey

Dear Dr. Lilian Nuwubaine,

Thank you for submitting your manuscript to PLOS Global Public Health. After careful consideration, there are still a few issues that need adjustment and modification. Therefore, we invite you to submit a revised version of the manuscript that addresses the points raised during the review process, as shared directly here.

EDITOR:

Carefully review any typos and address issues raised previously. For example, it has been asked to correct the following points in the last review, however it has not been done: Page 4, Line 89, remove the capital letter in "Reproductive age", it should be written with a lower case; Page 7, Line 147: After "These included", a colon should be used, not a semicolon.Specifically, in the abstract: Line 45, instead of "regional sensitive", replace with "region-sensitive".Page 3, Line 62: Stating "women's inferiority" connotes sexism. Women are not inferior to men, it is a patriachal and sexist society that marginalises women. Please rephrase your sentence. Same line: Remove the space after "inclusive".Please remove all contractions of verbs in the text, such as "don't play" or "couldn't" and write them in full terms/verbs.Page 3, Line 64: Please change the verb "indulge" with "undergo". Rephrase this sentence, including the part with " to avoid failing the test". Women and their partners resort to other forms of sexual relationships in fear of societal discrimination and consequences?Page 5, Line 108: Change "special" with "specific".Page 7, Line 152: after parity, you can add in parentheses "number of children" as asked by one of the reviewers.Page 17, Lines 284-286: Please add references to justify this argument. Line 285: Replace "highly" with "likely".Page 18, Lines 310-311: What are the other possible reasons, and add the references.Page 18, Lines 322-323: The article you chose is in contradiction with the previous sentence. Please review to ensure cohehence.Page 20, Line 354: Remove the space before "interventions".

We look forward to receiving your revised manuscript.

Kind regards,

Muriel Mac-Seing, PhD

Academic Editor

Journal Requirements:

2. We have noticed that you have uploaded Supporting Information files, but you have not included a list of legends. Please add a full list of legends for your Supporting Information files after the references list.

---

## [Editor Report · Decision Letter 3]

21 Nov 2023

PGPH-D-23-00992R3

Factors associated with HIV testing among pregnant women in Rwanda: a nationwide cross-sectional survey

Dear Dr. Nuwabaine,

Thank you for submitting your revised manuscript to PLOS Global Public Health. Please ensure you do find the passages of your manuscript where changes need to be made. You need to locate those according to the page and line specifications. Therefore, we invite you to submit a revised version of the manuscript that addresses the points raised during the review process.

EDITOR: 

Please address these two pending issues as shared with you in the previous review. Do provide references for the following passages of your third revised manuscript: Page 18, Lines 314-316; Page 18, Lines 316-318.For the following section (Page 18, Line 321 and Page 19, Lines 322-323), it is not necessarily a contradiction that both married women and women who have multiple partners have higher odds of HIV testing as married women might have more than one sexual partner and/or feel the importance of being tested for HIV, and women who have multiple partners may be single or not. Both groups of women are different, and for different reasons and contexts, they might be more likely to be tested for HIV. Please rephrase and indicate where changes were made.

We look forward to receiving your revised manuscript.

Kind regards,

Muriel Mac-Seing, PhD

Academic Editor
---

## [Editor Report · Decision Letter 4]

28 Nov 2023

Factors associated with HIV testing among pregnant women in Rwanda: a nationwide cross-sectional survey

PGPH-D-23-00992R4

Dear Dr. Nuwabaine,

We are pleased to inform you that your manuscript 'Factors associated with HIV testing among pregnant women in Rwanda: a nationwide cross-sectional survey' has been provisionally accepted for publication in PLOS Global Public Health.

Best regards,

Muriel Mac-Seing, PhD

Academic Editor